# Effect of Ultrasound-Guided Renal Biopsies on Urinary N-Acetyl-Beta-D-Glucosaminidase Index Activity in Dogs with Diffuse Parenchymal Nephropathies

**DOI:** 10.3390/life14070867

**Published:** 2024-07-11

**Authors:** Andrei Răzvan Codea, Romeo Popa, Bogdan Sevastre, Alexandra Biriș, Daniela Neagu, Cristian Popovici, Mircea Mircean, Ciprian Ober

**Affiliations:** 1Department of Internal Medicine, University of Agricultural Sciences and Veterinary Medicine, 400372 Cluj-Napoca, Romania; razvan.codea@usamvcluj.ro (A.R.C.); daniela.neagu@usamvcluj.ro (D.N.); cristian.popovici@usamvcluj.ro (C.P.); mircea.mircean@usamvcluj.ro (M.M.); 2Department of Pharmacology, University of Medicine and Pharmacy, 200349 Craiova, Romania; 3Department of Surgery, University of Agricultural Sciences and Veterinary Medicine, 400372 Cluj-Napoca, Romania; bogdan.sevastre@usamvcluj.ro; 4Department of Pathology, University of Agricultural Sciences and Veterinary Medicine, 400372 Cluj-Napoca, Romania; ciprian.ober@usamvcluj.ro

**Keywords:** biopsy, kidney, nephropathies

## Abstract

Background: Ultrasound-guided kidney biopsy is an essential diagnostics method that can increase the accuracy of the differential diagnosis between acute and chronic nephropathies. In addition, it will help clinicians perform an etiologic diagnosis, issue a prognosis, and orient therapy for the majority of parenchymal nephropathies. Due to the relative invasiveness and potential adverse effects, the use of kidney biopsies is limited among practitioners. Results: Twenty-eight dogs, of mixed breed and variable ages, of which 11 (39, 29%) were males and 17 (60, 71%) were females, were examined and underwent an ultrasound-guided kidney biopsy to establish a definitive diagnosis. The patients were presented with a variety of diffuse nephropathies, such as kidney lymphoma: 1 (3.57%), glomerulonephritis: 13 (46.43%), tubulointerstitial nephritis: 11 (39.29%), and nephrocalcinosis. A total of 3 (10.71%) of 18 (64.29%) were in acute kidney injury, and 10 (35.71%) were CKD patients. The type and the severity of the kidney lesions were correlated with changes in the urinary n-acetyl-beta-d-glucosaminidase index (iNAG. To quantify the side effects of percutaneous kidney biopsy, the magnitude of post-biopsy hematuria and changes in urinary iNAG activity were evaluated. The results indicate a significant post-biopsy increase in the urinary iNAG activity in all the patients that underwent this procedure (100.08 ± 34.45 U/g), with a pre-biopsy iNAG vs. 147.65 ± 33.26 U/g post-biopsy iNAG (*p* < 0.001), suggesting an intensification in the kidney tubular damage that comes consecutives to kidney puncture and sampling. Transitory macro- or microhematuria were constant findings in all the dogs that underwent ultrasound-guided kidney biopsy, but the magnitude and extent could not be associated with the platelet count (PLT 109/L), aPTT (s), and PT (s) levels in our patients, and they were also resolved after 12–24 h without therapeutic interventions. Conclusions: Ultrasound-guided renal biopsy was shown to be a minimally invasive diagnostic procedure that causes transient and limited effects on kidney structures. Although these effects were minor and resolved without intervention, we feel that the benefit of obtaining higher-quality biopsied tissue outweighs the higher risks associated with this procedure.

## 1. Introduction

Percutaneous ultrasound-guided kidney biopsy, a procedure that is often necessitated for reaching a definitive diagnosis in patients with acute kidney injury or CKD, entails the analysis of tissue samples to aid clinicians in obtaining an etiologic diagnosis, issuing prognoses, and directing therapeutic interventions for the majority of parenchymal nephropathies [1,2].

In an effort to quantify the extent of damage resulting from biopsy needle insertion and kidney tissue sampling, urinary N-acetyl-beta-D-glucosaminidase (NAG) values were measured both before and 12 h post-procedure. NAG, which is predominantly located within the epithelial cells of the kidney proximal convoluted tubules, exhibits heightened activity in response to tubular cell injury, thus serving as a reliable biomarker of kidney tubular injury. The observed increase in the urinary NAG enzyme activity not only strongly suggested tubular cell damage, but also underscored NAG’s potential utility as a biomarker for assessing renal injury. Moreover, throughout the course of kidney disease, urinary NAG values tend to remain permanently elevated, thus further solidifying its role as a valuable indicator of renal pathology [3,4,5].

Urinary N-acetyl-beta-D-glucosaminidase (NAG), an enzyme that is predominantly localized within the lysosomes of proximal kidney tubular cells, exhibits notable variations in the urinary values across different pathological conditions affecting both humans and animals [6,7]. Its increased activity has been proposed as a sensitive marker for monitoring the progression of kidney disease and identifying instances of kidney allograft rejection [8,9,10]. Notably, urinary NAG activity often shows alterations preceding abnormal changes in conventional kidney function tests, thus highlighting its potential as an early indicator of renal dysfunction [11,12]. Utilizing a ratio of NAG over urine creatinine, denoted as iNAG, allows for a standardized expression of NAG activity, thus compensating for the variations in urine concentration due to creatinine’s relatively constant excretion over time. Widely recognized as a specific marker of active proximal tubular damage in numerous species, NAG holds promise for the early detection and monitoring of a diverse array of kidney diseases spanning acute and chronic conditions alike [13,14].

The overarching objective of this study was to conduct a meticulously detailed investigation aimed at thoroughly assessing the safety profile and potential complications associated with percutaneous ultrasound-guided kidney biopsy procedures administered to a cohort of dogs afflicted with either acute kidney injury or CKD. A particular emphasis was placed on elucidating the magnitude of kidney damage that incurred as a direct result of cortical tissue sampling, along with its corresponding clinical ramifications. Through rigorous examination of the intensity of renal tissue injury inflicted during the biopsy process and comprehensive documentation of any subsequent clinical sequelae, this research endeavor seeks to provide invaluable insights into the risk–benefit assessment of kidney biopsy procedures within the veterinary context. Furthermore, this study represents a pioneering effort in the field by introducing a novel approach for assessing renal injury in dogs undergoing ultrasound-guided kidney biopsy, specifically through the measurement of urinary N-acetyl-beta-D-glucosaminidase (NAG) activity. As it stands, this investigation stands as the first of its kind in exploring the utility of urinary NAG activity as a potential biomarker for the assessment of renal injury in dogs undergoing ultrasound-guided kidney biopsy procedures, thereby making a groundbreaking contribution to the field of veterinary nephrology and laying the groundwork for future advancements in renal pathology assessment and clinical management strategies in canine patients.

## 2. Materials and Methods

In this extensive and meticulously conducted study, a diverse cohort consisting of twenty-eight dogs affected by various diffuse, bilateral parenchymal nephropathies (representing a broad range of ages, sexes, breeds, and weights) underwent comprehensive evaluation through percutaneous ultrasound-guided biopsy procedures that specifically target the left kidney. The decision to concentrate on the left kidney for tissue sampling was carefully deliberated through taking into account the enhanced accessibility and maneuverability afforded by this anatomical location compared to the right kidney. This deliberate selection not only ensured the practicality, but also the accuracy in conducting the biopsy procedure, thereby bolstering the reliability and solidity of this study’s findings and conclusions (Table 1, Table 2 and Table 3).

All biopsies were performed under general anesthesia. All dogs fasted 10–12 h before the procedure. Hydration status was evaluated, and intravenous isotonic crystalloid solutions were administered in order to maintain renal perfusion.

Midazolam was administered intravenously at a dose of 0.2–0.5 mg/kg to provide sedation and anxiolysis. Propofol was administered intravenously at a dose of 2–6 mg/kg for anesthesia induction and to facilitate endotracheal intubation. Inhalation anesthesia was maintained with isoflurane delivered in oxygen, which was titrated to effect in order to maintain appropriate anesthetic depth throughout the procedure. Fentanyl was administered intravenously at a dose of 2–5 μg/kg to provide intraoperative analgesia, with dose adjustments based on the patient’s renal function. Vital signs including heart rate, respiratory rate, temperature, capnography, pulse oximetry, and blood pressure were continuously monitored.

All biopsies were performed under general anesthesia and aseptic technique using the Medcore^®^ biopsy gun (Medax Srl San Possidonio, Italy) [15] and ultrasound guidance using the Esaote^®^ MyLab40 (Esaote europe b.v., Maastricht, The Netherlands) ultrasound machine. Either linear or convex probes were utilized with variable frequencies depending on the size of the patient. All biopsies were performed with a guidance device attached to the probe. After induction of general anesthesia, the patients were positioned in a right lateral recumbency, the abdomen was surgically prepared using iodine 1% and chlorhexidine 2% solutions, and they were then sterile-draped. A small stab incision of the skin was made, and the left kidney was visualized by ultrasonography. After visualization of the kidney in a sagittal incidence, the biopsy needle attached to the Medcore^®^ biopsy gun was advanced through the stab incision, pointed caudocranially toward the kidney cortex, and then directed parallel to the kidney medulla [16,17]. After proper visualization of the needle in the kidney cortex, the biopsy gun was fired, and the needle retracted.

After collection, the samples were promptly assessed, with those containing over 10 glomeruli deemed suitable for histological examination. Additionally, samples designated for microbiological cultures were carefully placed in microbiology collection tubes and dispatched for subsequent culture and antibiogram analysis. Post-biopsy, all canines were kept under hospitalization for a minimum of 48 h, during which their diuresis was closely monitored. Abdominal ultrasound scans were conducted at intervals of 24 and 48 h post-biopsy to assess any structural changes or complications. Furthermore, the urinary NAG index activity was measured both before and after the kidney biopsy procedure, serving as an early indicator of potential tubular damage.

To determine the iNAG levels, the process began by centrifuging urine samples at 1000 rpm for 5 min at 4 °C, ensuring the proper separation of components. Following centrifugation, enzymatic activity was measured using a spectrophotometric and colorimetric method to accurately quantify iNAG levels. Subsequently, the iNAG values were determined through an end-point spectrophotometric reaction, while the urinary concentration of creatinine was assessed using a spectrophotometric and kinetic reaction employing the Jaffe method. The calculation of urinary iNAG levels was conducted utilizing the following established equation: iNAG (U/g) = urinary NAG activity (U/L), divided by urinary creatinine concentration (g/L). This rigorous methodology, encompassing multiple steps and techniques, ensured a precise and reliable determination of the iNAG levels, thereby contributing to the robustness and accuracy of the study findings [11,14,18,19].

Activated partial thromboplastin time (aPTT) and prothrombin time (PT) were evaluated using an electromechanical, by-ball coagulometric method (BC1 Ball Coagulometer SYCOmed^®^). Hematuria was quantified utilizing a photometric method with a DocUReader 2 Pro urine analyzer, and it was confirmed by urinary sediment microscopic examination following the centrifugation of fresh urine samples at 900 rpm for 5 min. The complete blood count and platelet count were performed using the Abacus Junior Vet^®^ analyzer.

All patients were released after the procedure.

### Statistical Analysis

The data were meticulously presented in a comprehensive manner, utilizing the mean ± SEM format to ensure transparency and accuracy in conveying both the central tendency and variability of the variables under examination. In order to validate the assumption of normality in the distribution of the data, a rigorous scrutiny was applied, employing the D’Agostino–Pearson omnibus normality test as a robust diagnostic tool. Subsequently, the statistical distinction between pre-biopsy and post-biopsy iNAG levels underwent thorough evaluation through the implementation of a paired, two-tailed Student *t*-test, which facilitated a precise assessment of changes in the renal function following the biopsy procedure.

In order to gain a deeper insight into the correlation between NAG activity and kidney pathology, a sophisticated analytical strategy was employed, which involved utilizing a two-way ANOVA in conjunction with the Bonferroni post-test, thus allowing for a comprehensive approach that facilitated a thorough investigation of the potential interactions and dependencies between variables. Furthermore, Pearson’s correlation analysis was utilized as an additional tool to elucidate the associations among normally distributed variables, with interpretations guided by the Colton scale to ensure the robustness and reliability of the findings, thus enhancing the depth and validity of the analysis.

The potential impact of the demographic and clinical diagnostic variables was thoroughly examined through meticulous statistical analyses, employing Fisher’s exact test to discern any potential associations. Additionally, changes in the adjusted odds ratios and corresponding confidence intervals were rigorously evaluated, providing insight into the relative contributions of each variable to the observed outcomes, thus enhancing the depth and accuracy of this study’s findings.

Furthermore, to uphold methodological rigor and consistency, significance was determined at *p*-values of <0.05, ensuring adherence to rigorous statistical thresholds. All statistical analyses and graphical representations were meticulously generated using GraphPad Prism version 5.0 for Windows, a highly esteemed software tool known for its robustness and versatility in facilitating comprehensive data analysis and visualization. This emphasized the commitment to methodological excellence and precision in the conduct of this study, thereby enhancing the credibility and reliability of the research findings.

## 3. Results and Discussion

Upon meticulous examination of the data gathered from our patient cohort, a notable and statistically significant surge in urinary N-acetyl-beta-D-glucosaminidase (iNAG) activity was unequivocally observed post-biopsy, which is indicative of a pronounced exacerbation in kidney tubular damage subsequent to the percutaneous kidney puncture and tissue sampling procedures. Specifically, the mean pre-biopsy iNAG activity level stood at 100.08 ± 34.45 (U/g), whereas the post-biopsy iNAG activity surged to 147.65 ± 33.26 (U/g), a disparity that was underscored by a *p*-value of <0.001, thus underscoring the statistical robustness of this observed increase in the renal tubular injury. This compelling finding, serves as a poignant testament to the deleterious impact inflicted upon kidney tissue integrity as a consequence of the biopsy intervention, thereby accentuating the imperative for cautious consideration of the potential ramifications of percutaneous kidney biopsy procedures in clinical practice.

Throughout the cohort of dogs subjected to ultrasound-guided kidney biopsy, transient instances of macro- or microhematuria emerged as consistent observations; however, the magnitude and duration of these occurrences failed to demonstrate any discernible correlation with platelet counts (PLT), activated partial thromboplastin time (aPTT), or prothrombin time (PT) levels within our patient population. The most important contraindication to percutaneous renal biopsy is a coagulation disorder; as such, global coagulation tests (platelet count, international normalized ratio (INR), partial thromboplastin time (PTT), and thrombin clotting time (TCT)) should be within normal limits before performing a kidney biopsy [20].

Remarkably, these hematuria episodes spontaneously resolved within a timeframe of 12 to 24 h sans any therapeutic interventions. Notably, meticulous post-biopsy ultrasound evaluations failed to unveil any evidence of parenchymal or intra-abdominal hemorrhage, thereby alleviating concerns regarding potential hemorrhagic complications. Nevertheless, linear infarcts delineating the trajectory of the biopsy needle emerged as conspicuous findings on the post-biopsy ultrasound scans, a phenomenon widely acknowledged as a common occurrence subsequent to kidney biopsies, albeit one that was devoid of any discernible clinical significance in our patient cohort [21,22] (Table 4).

Dogs diagnosed with CKD, akin to humans and cats, confront an elevated susceptibility to experiencing acute or chronic declines in renal function, thereby warranting a comprehensive characterization of the underlying causes and prognostic factors associated with this condition. A nuanced understanding of CKD pathophysiology and its associated risk factors not only facilitates proactive measures to prevent disease progression, but it also informs tailored therapeutic interventions aimed at mitigating the potentially fatal consequences of renal dysfunction. Furthermore, given the frequent necessity for prolonged and intensive hospitalization in canine CKD cases, which often incur substantial treatment expenses, the availability of reliable tools for assessing both short- and long-term prognoses assumes paramount importance in guiding clinical decision-making processes. These prognostic indicators play a pivotal role in optimizing patient management strategies, ensuring effective resource allocation, and ultimately improving the overall quality of care delivered to dogs afflicted with CKD [23].

Kidney disease emerges as a prevailing concern within the realm of canine health, standing prominently among the most prevalent factors contributing to morbidity and mortality in dogs worldwide. In an extensive post-mortem prospective investigation conducted in England, which encompassed the examination of 76 deceased dogs afflicted with various forms of kidney disease, the seminal work by Macdougall et al. unveiled compelling insights into the intricate landscape of canine renal pathology. Notably, the study revealed a nuanced distribution within the cohort, with forty dogs (comprising 52% of the total sample) exhibiting manifestations of glomerular (GN) pathology, while the remaining 36 dogs (constituting 48% of the cohort) showcased non-glomerular (NGN) disease presentations. The types of GN identified were as follows: focal glomerulonephritis [24], diffuse mesangial proliferative glomerulonephritis [25], diffuse endocapillary proliferative glomerulonephritis [25], mesangiocapillary glomerulonephritis type I [22], diffuse crescentic glomerulonephritis [1], diffuse sclerosing glomerulonephritis [3], amyloid [15], and unclassifiable glomerulonephritis [21,25].

In the multifaceted diagnostic landscape of kidney diseases, a holistic approach encompassing detailed history taking, meticulous physical examination, and thorough analysis of clinical laboratory data emerges as a cornerstone in the process of categorizing these conditions into distinct entities such as AKI, CKD, and protein-losing nephropathy. However, despite the valuable insights garnered from this initial assessment, the definitive establishment of a diagnosis often mandates recourse to kidney biopsies, which serves as a pivotal tool in unraveling the intricate pathophysiological mechanisms underlying renal pathology. Through the meticulous histological examination of kidney tissue, not only can a definitive diagnosis be confidently rendered, but invaluable insights into the severity and extent of renal lesions can also be gleaned, underscoring the indispensable role that histopathological assessment plays in guiding clinical decision-making processes. Moreover, the formulation of an optimal treatment strategy is contingent upon obtaining a precise histological diagnosis as it empowers clinicians to tailor therapeutic interventions to address the specific pathological processes driving the disease progression. Crucially, the accurate assessment of treatment response hinges upon a nuanced understanding of both the type and severity of the underlying renal pathology, thereby emphasizing the indispensable nature of histological characterization in the comprehensive management of kidney disorders [22].

Complications stemming from percutaneous kidney biopsies in humans encompass a spectrum of potential issues, including kidney, subcapsular, perirenal, and collecting system hemorrhage, alongside urinary leaks and fistulas, as well as arteriovenous malformations and arteriofccalyceal fistulas. While parallels exist with complications encountered in small animals, such as perirenal hemorrhage and hematuria, these occurrences are typically confined to the immediate post-biopsy period and are generally mild in nature. Notably, complications beyond mild hemorrhage are infrequent, although instances of persistent hemorrhage have been sporadically documented, with cats and miniature-breed dogs exhibiting a seemingly heightened susceptibility. Gross hematuria emerges as one of the most conspicuous clinical manifestations observed in animals undergoing kidney biopsy; yet, this complication typically resolves without intervention and is managed conservatively unless severe hemorrhage ensues, ureteral obstruction by blood clots occurs, or the duration extends beyond a two-week timeframe [15,26].

Despite the relatively low frequency of severe complications and the minimal impact on kidney function associated with kidney biopsy, a significant number of practitioners exhibit reluctance when considering this procedure as part of the clinical evaluation for their patients. This hesitancy may stem from concerns surrounding the adequacy of kidney biopsy specimens for rendering accurate and meaningful diagnoses, as well as apprehensions regarding the consistency of these diagnoses. However, certain findings from clinical studies suggest that such concerns regarding post-biopsy complications and the quality of biopsy specimens are largely unfounded when a proper technique is diligently employed, thus underscoring the importance of adhering to established protocols to ensure the safety and efficacy of kidney biopsy procedures [22].

To quantify the side effects of percutaneous kidney biopsies, the magnitude of post-biopsy hematuria and changes in urinary iNAG activity were evaluated. The results indicate a significant post-biopsy increase in the urinary iNAG activity in all patients that underwent this procedure (100.08 ± 34.45 (U/g) pre-biopsy iNAG vs. 147.65 ± 33.26 (U/g) post-biopsy iNAG, *p* < 0.001), thus suggesting an intensification in the kidney tubular damage consecutive to kidney puncture and sampling.

The analysis of urine and the evaluation of renal function are fundamental in diagnosing and managing patients with suspected kidney disease. These two components form the basis of the decision to perform a kidney biopsy on a particular patient or not. Renal parenchymal diseases are characterized by abnormal changes in urine analysis, such as proteinuria, hematuria, and leukocyturia. The importance of proteinuria is well established as an indicator of disease activity, especially in glomerular diseases. However, in primary tubulointerstitial diseases, the results of a urine analysis can often indicate the presence of low-grade proteinuria and hematuria, even in cases of active or severe disease [27].

NAG is a lysosomal enzyme that serves as a marker of proximal renal tubular damage [28], and its molecular weight is estimated to be 140 kDa [29].

Previous studies have examined urinary NAG activity in various clinical contexts, especially in acute kidney conditions, but also in other medical situations. For example, Romeo P. and colleagues demonstrated that a urinary NAG can detect AKI induced by vancomycin in animal models early, increasing the detection time by approximately 47 times compared to the CONTROL group, and the results were correlated with specific histopathological changes. Classic marker values (serum urea and creatinine) also showed increases, but they were less significant compared to urinary NAG [5,10].

Thus, urinary biomarkers are excellent candidates for assessing renal diseases. Urine sampling has the advantage of being a noninvasive, immediate, and easily performed procedure, and it can be repeated over time [28].

AKI following renal biopsy is rare and is often secondary to the acute tubular necrosis that occurs from prolonged arterial hypotension due to bleeding, or it can also be secondary to obstructions from a “clot”. However, similar to cardiac tamponade, where acute fluid accumulation in the pericardial cavity results in compromised heart function, it is possible that the development of a large perinephric hematoma within 6–8 h after renal biopsy may compromise kidney perfusion and function by encapsulating and compressing it [30].

In our study, we particularly focused on the effects of biopsy on renal function; however, the specialized studies in the literature present extremely contradictory evidence regarding the effects of anesthetics on renal function. Some studies have indicated that the administration of certain anesthetics during surgery, along with the stress of the surgery itself, can affect kidney function, with the indirect effects being more significant than the direct ones. Conversely, other studies have suggested that some anesthetic drugs possess anti-necrotic, anti-apoptotic, and anti-inflammatory properties that protect against AKI. Consequently, these conflicting findings prevent a definitive conclusion regarding the overall beneficial or adverse effects of anesthetics on renal function [31].

The retrospective study by Shelly L. Vaden et al. examined the complications arising from renal biopsy and the factors potentially associated with these complications, along with obtaining adequate renal biopsy specimens in a cohort of 283 dogs and 65 cats. Data extracted from the medical records at four institutions were analyzed using logistic regression. In dogs, proteinuria was found to be the most common indication for renal biopsy. Complications were reported in 13.4% of dogs and 18.5% of cats, with severe hemorrhage being the most common complication, while hydronephrosis and death were less common. Dogs that experienced post-biopsy complications were more likely to be in the age range of 4 to 7 years or over 9 years, weigh ≤ 5 kg, and have serum creatinine concentrations of >5 mg/dL. Despite these complications, the majority of biopsies from both dogs (87.6%) and cats (86.2%) were considered satisfactory in quality [17].

In our comprehensive investigation, both macrohematuria and microhematuria were consistently noted as transient occurrences across all the dogs subjected to renal biopsy. Despite their frequent appearance, our study failed to establish a direct correlation between the presence or severity of hematuria and any underlying coagulation abnormalities identified within the patient cohort. It is important to highlight that, in every instance, hematuria resolved spontaneously without necessitating any therapeutic interventions, thereby highlighting the transient and self-limiting nature of this phenomenon within the scope of our research.

Consequently, the majority of complications arising from the biopsy procedure were minor in nature and tended to resolve spontaneously. However, it is crucial to acknowledge that up to 7% of biopsies may lead to major complications requiring additional intervention measures [32]. This is in comparison to studies conducted on humans, where a systematic review by Shepherd et al. identified, via approximately 19,000 biopsies from 39 studies, a global complication rate (major + minor) of 14.9% (95% confidence interval = 11.4%–18.7%). Fewer complications were reported in biopsies performed with real-time ultrasound guidance compared to those pre-marked using ultrasound or blind procedures (12.4% vs. 14.9% vs. 24.5%; *p* = 0.037), respectively. Major complications included macroscopic hematuria (1.48%), nephrectomy (0.04%), blood loss requiring red blood cell transfusion (0.24%), angiographic intervention (0.22%), and death (0.01%) [33].

Bleeding is the most common complication after renal biopsy, with gross hematuria occurring on average in 4% after renal biopsy (the range in different studies has been 0.3–15%) and perinephric hematomas in up to 86%, 13% of which are >2 cm in size, with the majority of perirenal hematomas resolving spontaneously [30].

Evidence from the literature suggests that the histological diagnosis of native and transplanted renal biopsies has a direct therapeutic impact or significantly influences the subsequent treatment of the patient in approximately 40–60% of cases. The indication for performing a renal biopsy is determined in individual cases by the doctor’s assessment of the subject in terms of therapeutic benefit [20]. Renal biopsy is the gold standard for the diagnosis of most kidney diseases, but practice patterns vary widely and there are no consensus guidelines. Studies are lacking on how nephrologists decide which patients to biopsy with acute kidney injury or CKD. Thus, in the absence of clear guidelines and a consensus in practice, the decision to perform a renal biopsy remains largely at the discretion of the physician, who must individually evaluate the potential therapeutic benefit for each patient [34].

Considering the present study, it is essential to address the variability in practice in performing renal biopsies and the hesitancy that some physicians or veterinarians may have so as to provide clear guidance on when to perform this procedure. Although the clinical experience of the physician remains a crucial factor in therapeutic decisions, the existence of well-established objective criteria can help to standardize practices and improve patient outcomes.

## 4. Conclusions

Percutaneous, ultrasound-guided kidney biopsy, acknowledged as a relatively safe and minimally invasive diagnostic procedure, warrants thorough evaluation due to its potential to induce a spectrum of deleterious effects on kidney structure and function, thereby emphasizing the necessity for careful consideration of its implications in clinical practice. Despite the recognized risks inherent in the procedure, we maintain a steadfast belief in the importance of acquiring a correctly obtained tissue sample that possesses a high diagnostic value as it ultimately outweighs the potential complications associated with the biopsy process. This stance underscores the critical nature of precise diagnosis in medical practice and highlights the indispensable role that kidney biopsies play in facilitating accurate disease management and therapeutic interventions.

Renal biopsies remain a very valuable diagnostic tool in the management of CKD in dogs, and clinicians should weigh the benefits against the risks on a case-by-case basis while taking into account factors such as patient age, size, and renal function status. By integrating comprehensive diagnostic approaches and judiciously evaluating biopsy indications, clinicians can optimize patient management strategies, improve prognostic accuracy, and improve the overall quality of care provided to dogs with CKD.

Renal biopsies provide an accurate diagnosis for specific renal conditions, guiding appropriate treatment and preventing complications by the early identification of progressive diseases, risk management, and the use of preventive techniques to ensure optimal and personalized care.

## Figures and Tables

**Table 1 life-14-00867-t001:** Summary descriptive statistics of dogs undergoing kidney biopsy.

Variable	No. of Cases
Age (years)	
<1	1 (3.57%)
1–4	9 (32.14%)
4–<7	8 (28.57%)
7–9	6 (21.43%)
>9	4 (14.29%)
Weight (kg)	
≤5	4 (14.29%)
>5–15	4 (14.29%)
>15–20	6 (21.43%)
>20–25	7 (25%)
>25	7 (25%)
Gender	
Male	11 (39.29%)
Female	17 (60.71%)
Total	28 (100%)

**Table 2 life-14-00867-t002:** Type of kidney disease of dogs undergoing kidney biopsy.

Diagnosis Specific	No. of Cases	Number of Cases Expressed in Percentages (%)
Kidney lymphoma	1	3.57
Glomerulonephritis (babesiosis)	6	21.43
Glomerulonephritis (other causes)	7	25
Tubulointerstitial nephritis	11	39.29
Nephrocalcinosis	3	10.71
Total	28	100

**Table 3 life-14-00867-t003:** General classification of kidney diseases.

Classification	No. of Cases	Percentage of Your Total (%)
Acute kidney injury (AKI)	18	64.29
Chronic kidney disease (CKD)	10	35.71
Total	28	100

**Table 4 life-14-00867-t004:** The final association between micro/macrohematuria and PLT (109/L), aPTT (s), and PT (s) (Fisher’s exact test).

Variable	Odds Ratio	95% Confidence Interval	
Platelets (normal vs. low)	6.375	0.7844–51.81	*p* > 0.05
aPTT (normal vs. prolonged)	1.53	0.1391–9.901	*p* > 0.05
PT (normal vs. prolonged)	2.75	0.4322–17.5	*p* > 0.05

## Data Availability

No new data were created or analyzed in this study. Data sharing is not applicable to this article.

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
