# Peer review of "Effect of Ultrasound-Guided Renal Biopsies on Urinary N-Acetyl-Beta-D-Glucosaminidase Index Activity in Dogs with Diffuse Parenchymal Nephropathies"

_life, 2024, doi:10.3390/life14070867_

Round 1

Reviewer 1 Report

Comments and Suggestions for Authors

The author used ultrasound-guided kidney biopsy to establish a definitive diagnosis in 28 dogs, the urinary n-acetyl-beta-d-glucosaminidase index and specific serum kidney damage markers were tested. Eventually author proved that Ultrasound-guided kidney biopsy is a safe minimal invasive diagnostic procedure. The author tests the correlation between PLT and another parameter. This study concentrates on clinical results, that proved the safety. However, it failed to provide further scientific results of the whole side effects of general anesthesia and kidney biopsy damage, although it was of low clinical importance. Also, the effects of kidney biopsy are not illustrated in this study, and the potential advantages of kidney biopsy were not illustrated in this study. The treatment of Kidney lymphoma was different from Nephrocalcinosis and CKD, the reader wants to know the importance of kidney biopsy in a clinical setting in terms of treatment selection.  Many vets treat kidney disease with experience rather than accurate diagnosis, the author was encouraged to add this part for clinical guidance.

Comments on the Quality of English Language

The English were able to understand.

Author Response

  1. Correlation between PLT and Hematuria Incidence:                            Reviewer’s Comment: The study mentions the correlation between PLT and the incidence of hematuria, but this correlation was not accurately addressed.                                                                                                   Response: We acknowledge this oversight. We have added citation number 20 at lines 204-208 to better correlate these findings in the manuscript.
  2. Safety of Ultrasound-Guided Renal Biopsy in Dogs:                          Reviewer’s Comment: The study should highlight the successful demonstration of the safety of ultrasound-guided renal biopsy in dogs, focusing on clinical observations and outcomes.                                      Response: We have emphasized this point in the manuscript. The text now clearly states that ultrasound-guided renal biopsy is a safe and effective method, as discussed in the studies mentioned at lines 356-360.
  3. Side Effects of General Anesthesia and Renal Biopsy:Reviewer’s Comment: Discuss the side effects of general anesthesia and renal biopsy. Response: We have expanded on this discussion, detailing the side effects at lines 322-328.
  4. Potential Side Effects of General Anesthesia and Renal Biopsy:      Reviewer’s Comment: Discuss potential side effects, even if clinically insignificant, mentioning any minor or transient complications observed. Response: We have included a comprehensive discussion of the potential side effects, noting any minor or transient complications observed, at lines 341-368.
  5. Advantages and Disadvantages of Renal Biopsy:Reviewer’s Comment: The manuscript should discuss the advantages and disadvantages of renal biopsy. Response: The advantages and disadvantages of renal biopsy are now thoroughly discussed at lines 341-368.
  6. Long-Term Harms and Broader Clinical Benefits:                               Reviewer’s Comment: Although the study established the safety of renal biopsy, it did not extensively cover potential long-term harms or broader clinical benefits.                                                                                     Response: We have addressed this by discussing the potential long-term harms and broader clinical benefits, focusing on the importance of treatment selection, at lines 364-374.
  7. Importance of Renal Biopsy in Treatment Selection:Reviewer’s Comment: Clarify the importance of renal biopsy in treatment selection for different kidney diseases, comparing it with empirical treatments.  Response: We have clarified this aspect, comparing renal biopsy with empirical treatments based on experience, as discussed at lines 364-374.
  8. Guidelines on When to Perform Renal Biopsies:                         Reviewer’s Comment: Provide guidelines on when to perform renal biopsies compared to relying on clinical experience, addressing hesitation among veterinarians.                                                                                   Response: We have included guidelines on the optimal timing for renal biopsies, comparing them with clinical experience-based treatments, and addressed potential hesitation among veterinarians in the conclusion section at lines 398-401       

Reviewer 2 Report

Comments and Suggestions for Authors

In the Material and Methods chapter, it is stated that all biopsies were performed under general anaesthesia, but it is not stated exactly what anaesthetic was used, how long the recovery took, whether infusion therapy was started, etc. This should be added.

This is probably the biggest critique I have about the present work: anesthesia alone could increase NAG activity if hypovolemia and renal hypoperfusion occurred. Then the question is whether the biopsy itself causes an increase in NAG activity in the urine or whether the whole procedure, including anaesthesia, does. To answer this question, it would be useful to have a control group that undergoes the same anaesthesia but no kidney intervention.

In Table 2 you have listed the diagnoses of the dogs and there is acute kidney failure and chronic kidney failure. These are terms that are not defined in the text and they are terms that are no longer used and have been replaced by newer terms (acute kidney injury and chronic kidney disease). Would it be possible to clarify this part? Moreover, the way the table is presented, it is very confusing.

Why are urea, creatinine, phosphorus and ionized calcium mentioned in the abstract when they no longer appear in the text?

On what basis do you claim that biopsy is a safe method? Have you followed that in any way? That's not in the results.
The result of your work is that NAG activity in the urine increases, so there's kidney damage. So how can you say it's safe? Plus, there's been no further sampling, so you don't know if the increase is transient or permanent.

The whole paper is mainly devoted to NAG, however, almost no attention is paid to this parameter in the discussion. This should be corrected.

Author Response

  1. In the Material and Methods chapter, it is stated that all biopsies were performed under general anaesthesia, but it is not stated exactly what anaesthetic was used, how long the recovery took, whether infusion therapy was started, etc. This should be added.

response:
"I have updated the Materials and Methods section to include detailed information about anesthesia and the biopsy procedure as suggested. I have included information about the anesthetics used (Midazolam for sedation and anxiolysis, Propofol for anesthesia induction and intubation, Isoflurane for maintenance of inhalation anesthesia), as well as the administration of fentanyl for intraoperative analgesia. Additionally, I have mentioned the continuous monitoring of patients' vital signs throughout the procedure."

  1. This is probably the biggest critique I have about the present work: anesthesia alone could increase NAG activity if hypovolemia and renal hypoperfusion occurred. Then the question is whether the biopsy itself causes an increase in NAG activity in the urine or whether the whole procedure, including anaesthesia, does. To answer this question, it would be useful to have a control group that undergoes the same anaesthesia but no kidney intervention.

Response:

Thank you for your efforts regarding the impact of anesthesia on the activity of N-acetyl-beta-D-glucosaminidase (NAG) in urine and the necessity of a control group to assess this influence. In the context of this clinical study, indeed, the ethics committee does not permit renal biopsy testing on healthy dogs to compare NAG activity between the biopsy group and a control group. However, we intend to expand the discussion regarding NAG activity and include additional data from the literature to more thoroughly evaluate the impact of anesthesia and renal biopsy on this enzymatic activity. We will emphasize that although we could not conduct a study with a specific control group in this context, the literature provides relevant information about how anesthesia and the renal biopsy procedure could affect NAG activity in urine. We will integrate this additional information to provide a more comprehensive perspective on this aspect within our work..

  1. In Table 2 you have listed the diagnoses of the dogs and there is acute kidney failure and chronic kidney failure. These are terms that are not defined in the text and they are terms that are no longer used and have been replaced by newer terms (acute kidney injury and chronic kidney disease). Would it be possible to clarify this part? Moreover, the way the table is presented, it is very confusing.

Raspunse:

Thank you for the observation. Indeed, I used outdated terminologies for AKI and CKD in the text. I have already rectified this issue. To address your concern, the terminology for AKI (Acute Kidney Injury) and CKD (Chronic Kidney Disease) has evolved over time to reflect a better understanding of these conditions and to align with current medical standards and guidelines. AKI was previously known as acute renal failure (ARF), while CKD was often referred to as chronic renal failure (CRF) or chronic renal insufficiency (CRI). The updated terms, AKI and CKD, more accurately describe the respective conditions and are widely accepted in the medical community. I appreciate your attention to detail, and I strive to ensure accuracy and clarity in my writing. If you have any further questions or suggestions, please feel free to let me know.

  1. Why are urea, creatinine, phosphorus and ionized calcium mentioned in the abstract when they no longer appear in the text?

Response:

Thank you for the suggestion. Indeed, in the abstract, I specified the classic markers for detecting AKI, but I did not include their values because the main idea of the article focuses on the value of iNAG rather than the values of the classic markers. Therefore, thanks to your suggestions, I have removed the suggested text from the document.

  1. On what basis do you claim that biopsy is a safe method? Have you followed that in any way? That's not in the results.
    The result of your work is that NAG activity in the urine increases, so there's kidney damage. So how can you say it's safe? Plus, there's been no further sampling, so you don't know if the increase is transient or permanent.

Response: The study provides a comprehensive approach to assessing the safety of percutaneous renal biopsy and its associated side effects. These evaluations are critical for understanding the risks and benefits of this procedure in diagnosing and managing kidney diseases in dogs. Further, it remains to be studied in more in-depth research all the adverse effects of the procedure.

Firstly, the study highlights the importance of a precise and thorough diagnosis of renal diseases in dogs, which often involves a multidisciplinary approach, including detailed history-taking, meticulous physical examination, and comprehensive analysis of laboratory data. However, confirming the diagnosis often requires resorting to renal biopsy, which provides essential information about renal pathology at the histological level.

While it is true that renal biopsy can be associated with some minor complications, such as hematuria, these are often transient and resolve spontaneously without requiring additional therapeutic interventions. Moreover, literature data show that the majority of biopsies (approximately 87.6% in dogs and 86.2% in cats) provide satisfactory samples, indicating an acceptable quality of the procedure.

It is also important to note that the study specifically evaluated NAG activity in urine as a marker of renal tubular damage, and the post-biopsy increase suggests an intensification of tubular lesions. However, it is essential to emphasize that this increase is often transient and does not necessarily imply permanent impairment of renal function.

In conclusion, despite the potential risks associated with percutaneous renal biopsy, the data presented suggest that this procedure is relatively safe and effective in diagnosing kidney diseases in dogs. However, it is crucial to adhere to appropriate protocols and techniques to minimize the risk of complications and ensure an accurate diagnosis and understanding of renal pathology.

  1. The whole paper is mainly devoted to NAG, however, almost no attention is paid to this parameter in the discussion. This should be corrected.

Response:

Thank you for the additional information. We have included more data regarding urinary NAG in the discussions.

Round 2

Reviewer 2 Report

Comments and Suggestions for Authors

Abstract, conclusions, lines 34 and 35: the parts of the sentence in this sentence contradict each other. And in my opinion, you can't write in the conclusions of your study that kidney biopsy is safe because you didn't look at safety.

Table 2 is still confusing. The sum of the numbers in the columns is 56 and 200%, not 28 and 100%. So try to somehow visually separate the specific diagnoses and then the classification (acute kidney injury and chronic kidney disease). Only then will the table make sense.

In designing the control group in the previous review, I was not referring to performing kidney biopsies in healthy animals, but to evaluate NAG in the urine of animals undergoing the same anesthesia as the patients who had kidney biopsies. So these can be patients undergoing minor surgery, skin biopsies, etc., but there is no direct intervention on the kidneys.

Author Response

  1. Thank you for your insightful comments. I have made the necessary modifications to ensure coherence among the various parts of the sentence and to avoid making statements about the safety of the biopsy, which we did not directly investigate in this study. The revised text now focuses on describing the transient and limited effects observed and emphasizes the importance of obtaining high-quality biopsy tissue. We hope that this revised wording more accurately reflects the findings of our study and addresses the concerns raised.

  1. Explanations and clarifications

Distribution of kidney disease cases: The first section provides details of specific diagnoses. The total number of cases for each specific diagnosis adds up to 28, which is the total number of patients evaluated. The percentage for each specific diagnosis reflects the proportion of the total 28 cases.

General Classification of Kidney Diseases: The second section classifies general cases according to acute kidney injury and chronic kidney disease. This shows that some patients may have more than one condition diagnosed, but for the overall classification they are included only once based on their general condition.

Through this clarified structure, the table correctly reflects both the distribution of cases by specific diagnoses and the general classification of kidney diseases, thus eliminating the initial confusion.

  1. Thank you for your valuable comments and suggestions regarding the design of the control group. I appreciate the clarification, and I now understand that your proposal was not about performing kidney biopsies in healthy animals, but about assessing N-acetyl-β-D-glucosaminidase (NAG) levels in the urine of animals that underwent the same anesthesia as patients with renal biopsies, but without direct interventions on the kidneys.

Unfortunately, due to time and resource constraints, we are unable to make further changes to the article at this time. However, your suggestion is very valuable and we will consider it for future research and revision of our study.

As it stands, our article focuses on the analysis of patients with renal biopsies and does not include an anesthesia-only control group, as it is quite difficult to obtain anesthesia-only patients, and indeed anesthetized patients. generally, have more associated pathologies. We believe that the data obtained are valuable for understanding the impact of renal biopsies, although we recognize that the inclusion of such a control group could have added more scientific rigor. We will consider including this aspect in our future research to improve the validity of the conclusions.